# FoGE: Fock Space inspired encoding for graph prompting

## Abstract

Recent results show that modern Large Language Models (LLM) are indeed capable of understanding and answering questions about structured data such as graphs. This new paradigm can lead to solutions that require less supervision while, at the same time, providing a model that can generalize and answer questions beyond the training labels. Existing proposals often use some description of the graph to create an "augmented" prompt fed to the LLM. For a chosen class of graphs, if a well-tailored graph encoder is deployed to play together with a pre-trained LLM, the model can answer graph-related questions well. Existing solutions to graph-based prompts range from graph serialization to graph transformers. In this work, we show that the use of a parameter-free graph encoder based on Fock space representations, a concept borrowed from mathematical physics, is remarkably versatile in this problem setting. The simple construction, inherited directly from the theory with a few small adjustments, can provide rich and informative graph encodings, for a wide range of different graphs. We investigate the use of this idea for prefix-tuned prompts leveraging the capabilities of a pre-trained, frozen LLM. The modifications lead to a model that can answer graph-related questions – from simple graphs to proteins to hypergraphs – effectively and with minimal, if any, adjustments to the architecture. Our work significantly simplifies existing solutions and generalizes well to multiple different graph-based structures effortlessly.

## 1 Introduction

Large Language Models (LLMs) excel at tasks like question answering, sentence completion, translation, and even solving undergraduate-level math problems (Liu et al., 2024; Johansson, 2024). However, they sometimes need additional data unavailable during training. For instance, a model trained on data up to a specific date may struggle with the ever-changing news cycle (Vu et al., 2023; Mousavi et al., 2024). To prevent responses from becoming outdated, or to integrate non-public/proprietary data and domain-specific terminology, models need extra context. Retrieval Augmented Generation (RAG) describes this process of retrieving and integrating extra information to an LLM during its generation process. While multiple different approaches have been proposed for the retrieval part, the most common solution to the integration of the additional information is In-Context Learning (ICL) (Guu et al., 2020; Ding et al., 2024; Dong et al., 2022; Zoph et al., 2022; Min et al., 2022). ICL allows additional information to be included with a prompt, guiding the model to generate responses aligned with the extra context. This method is beneficial as it does not require retraining the LLM and can be applied to proprietary models like GPT (Brown et al., 2020) by adding a text description of the extra information.

ICL-type ideas are also being studied for utilizing not just additional/new data but also novel input formats/modalities, such as tables and graphs (Sui et al., 2024; Lu et al., 2024; Wang et al., 2023; Guo et al., 2023). While specialized models may still perform better at specific tasks, LLMs can serve as general-purpose reasoning machines, capable of answering questions about the provided modality beyond the training labels. Several recent results have reported success at "serializing" such structured data-types into a text-form description that can be easily used within ICL. For tables, the serialization is not too complicated (Sui et al., 2024; Lu et al., 2024), but more care is needed for graphs. While different types of graphs can all be handled by the same pipeline, the efficacy of the overall model varies from one setting to the other (Fatemi et al., 2024; Wang et al., 2023; Guo et al., 2023). Further, it has been observed that specific design choices to "textify" the graph can influence

performance and additionally, prompting techniques can have more than a small impact on the results (Fatemi et al., 2024). What will work well in a specific setting depends on both the question at hand as well as the characteristics of the data (Perozzi et al., 2024; Chai et al., 2023).

**Prefix-tuning.** One option to address the mentioned issues is "prefix-tuning" (Li & Liang, 2021). A specialized graph encoder translates the underlying graph into embeddings that can be fed directly to an LLM, eliminating the need for a textual description. Although not training-free, the LLM remains *frozen*, and only the *relatively smaller* graph encoder is trained. This approach has shown impressive performance, often surpassing ICL-based methods Sun et al. (2022); Liu et al. (2023); Tang et al. (2024). However, using a specialized graph encoder can be challenging due to the variety of graph types, and multiple works have proposed modifications of GNNs that suit their demands. For example, GraphToken (Perozzi et al., 2024) can encode only simple graphs, while GNP (Tian et al., 2024) constructs a complex pipeline to handle large graphs and extract subgraphs. GraphLLM (Chai et al., 2023) combines a transformer and a GNN (about $100M$ parameters), requiring detailed text descriptions for each node. Despite sophisticated designs, adapting these models to different graph types (e.g., protein-derived graphs or hypergraphs) is difficult, and even familiar graph types need adjustments for new tasks.

**Context of this paper.** ICL-based approaches for graphs primarily involve converting graphs to text, while prefix-tuning with graphs uses modules to extract richer, *task-relevant* structures, requiring larger sample sizes and higher compute power. A key question is whether we can achieve powerful, task-agnostic graph representations that are as easy to obtain as ICL-based methods. Could a lightweight adapter map these rich (but task-independent) representations into the LLM embedding space, making prefix-tuning effective for various tasks? Recent results hint that this may be viable (Moayeri et al., 2023). For instance, a *single linear layer* can transform an arbitrary image encoder's outputs to align with CLIP's (Radford et al., 2021) text encoder embeddings. If our graph encoding captures the graph's information and structure well enough, a similar adapter could work with a pre-trained LLM to offer good performance. This approach's success depends on

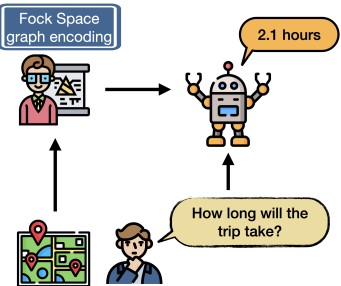

Figure 1: Augmenting LLM's capabilities by prompting them with carefully encoded graphs.

the quality of the graph representations. We ensure this by invoking a mature concept from mathematical physics, called Fock Spaces, whose practical instantiation yields almost lossless task-agnostic graph embeddings. Our findings show that a linear adapter with these representations yields competitive performance, handling complex graph questions and diverse structures like hypergraphs and proteins. The **main contribution** of this paper is the Fock-space inspired encoding of diverse graph-based structures, ranging from simple graphs to those obtained from proteins. We provide open-source code for grounding LLMs using our graph encodings as prompts and carefully profile the performance of this pipeline relative to baselines, on diverse datasets.

## 2 DERIVING FOCK SPACE BASED GRAPH REPRESENTATIONS

We will first review a few notations and results which will together provide the conceptual pipeline for obtaining our representations of graphs for prompting. While graphs serve as representative examples here, the rationale for other types of structured data such as tables is similar.

**Setup/rationale.** Consider a graph $G = (V, E)$ with a vertex set $V$ and an edge set $E$; $|\cdot|$ denotes set cardinality. We define the *incidence matrix* (Hatcher, 2002), $I$ to be of shape $|V| \times |E|$ where $I_{ij} = 1$ if *edge $j$ ends at vertex $i$*, $-1$ if *edge $j$ starts at vertex $i$* and 0 elsewhere. Let $|V| = n$. It is common to represent graphs via graph spectra derived from the Laplacian's eigenvalues. This is effective for studying global properties of graphs like connectivity/symmetries (e.g., Courant Fischer theorem, Fiedler's theorem (Fiedler, 1973; 1989)) but less so for capturing localized relationships between individual entities (nodes, edges, faces) within the graph. It turns out that an interesting direction using Clifford Algebra, shown so far to be effective in geometric problems in machine learning (Ruhe et al., 2023b; Chen et al., 2024; Ruhe et al., 2023b;a; Brehmer et al., 2023), provides rigorous tools for representing various graph elements (nodes, edges, faces) in a nice algebraic

structure Oziewicz (1998). Graphs can be embedded and manipulated in a geometric space Baylis (2012), and in principle, their spectral properties can also be studied. We briefly summarize the concept to lay out its benefits and challenges.

## 2.1 Clifford Algebra and Graph Representations

**Clifford Algebra.** Let $K$ be a field, i.e., comprised of elements that can be added, subtracted, multiplied, and divided (except by zero). Let $W$ be a $K$-vector space, i.e., it is a vector space over the field $K$ meaning that the vectors in $W$ can be added, subtracted, and multiplied by scalars (elements from $K$) following some rules. Let $W$ be equipped with a symmetric bilinear form $\langle \cdot, \cdot \rangle$ (or more generally, a quadratic form) where in case of graphs, $W = \mathbb{C}$, and let $T(W)$ denote the exterior algebra of $W$, a structure on *top* of the vector space $W$ to capture *all possible products* of vectors $w \in W$ (this will include scalar multiples and sums of products). Let $I(W)$ be the ideal in $T(W)$ generated by the set $\{w \otimes w + \langle w, w \rangle \mathbf{1}\}$, where $\mathbf{1}$ denotes the multiplicative identity in $K$ and $w \otimes w$ represents the product of a vector $w$ with itself in the exterior algebra. Recall that $I(W)$ is a subset of $T(W)$, with the property that the product of an element from $I(W)$ and an element from $T(W)$ is in $I(W)$ (a closure property). For an in-depth analysis, we point the reader to Dorst et al. (2009); Lounesto (2001).

**Definition 1.** *Let $W$ be a vector space over a field $K$, equipped with a quadratic form $q : W \to K$. The **Clifford algebra** of $(W, q)$, denoted $\mathfrak{Cl}(W, q)$, is the quotient algebra $T(W)/I(W, q)$.*

We take the exterior algebra $T(W)$ and "divide" it by the ideal $I(W)$. The ideal acts as a "filter" to filter out information captured by the ideal (all terms where a vector gets multiplied by itself along with its corresponding scalar term from the bilinear form) since they do not add much to the structure (set to scalar multiples of the identity element). But the ideal does more: it establishes an equivalence relation that changes the multiplication operation.

**One choice of Clifford Algebra representation.** A $K$-vector space allows scalar multiplication, while a $K$-algebra extends this with element-wise products. A representation of a $K$-algebra is a homomorphism $\rho$ that maps elements from the algebra to a vector space. For a $K$-vector space $W$, let $Hom(W, W)$ be the set of linear maps from $W$ to itself. Given a $K$-algebra $A$, we can define a homomorphism $\rho : A \to Hom(W, W)$. In the case where $A$ is the Clifford algebra $\mathfrak{Cl}_n(\mathbb{C}, q)$, $\rho$ allows representing its elements as linear operators on $W$, making it possible to manipulate these elements of $\mathfrak{Cl}_n(\mathbb{C}, q)$ concretely.

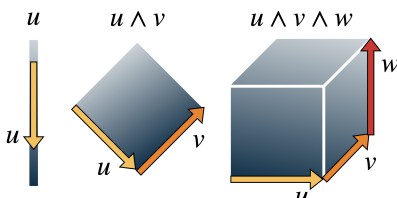

Figure 2: Single, Bi- and Tri-vectors in Clifford Algebra with wedge products.

**Practical Considerations in Clifford Algebra Operations.** We can closely follow the axioms of Clifford Algebra and through its wedge product build higher-order elements while preserving the geometric structure at hand (e.g., hyperedges or faces to multi-vectors). While cleanly rooted in theory, this leads to problems in practice. Implementing the full Clifford algebra structure over an $n$-dimensional vector space implies working with an algebra of dimension $2^n$. We must also define the multiplicative and graded structure, and, despite progress, software support is limited for higher dimensions (see (Zhdanov et al., 2024) and projects like Grassmann.jl and GeometricAlgebra.jl). Hence, we will need to make some design choices that balance practicality and mathematical soundness.

## 2.2 From Graphs to Clifford Algebra to Fock Spaces

**Dirac operator.** For graph $G$, we define the *Graph Laplacian* as $\Delta = II^T \in \mathbb{R}^{|V| \times |V|}$, where $I$ is the incidence matrix of $G$ (Casiday et al., 2024). Given the spectral decomposition $\Delta = Q\Lambda Q^T$, where $Q \in \mathbb{R}^{|V| \times |V|}$ is orthogonal and $\Lambda = \text{diag}(\lambda_1, \cdots, \lambda_{|V|})$ is diagonal with eigenvalues $\lambda_i \geq 0$, we define the *Dirac operator* as: $D = Q\sqrt{\Lambda}Q^T \in \mathbb{R}^{|V| \times |V|}$. Let $\{e_1, \cdots, e_{|V|}\}$ be the standard basis for $\mathbb{R}^{|V|}$. We can express $D$ in terms of a finite basis expansion: $D = \sum_{k=1}^{|V|} E_k \otimes \frac{\partial}{\partial e_k}$. Here, $\frac{\partial}{\partial e_k}$ are partial differential operators corresponding to directions $e_k$. The coefficient matrices $E_k \in \mathbb{R}^{|V| \times |V|}$

gives the action of $D$ in each coordinate direction. Specifically, we can say $E_k = D \cdot \text{diag}(e_k)$ where $\text{diag}(e_k)$ is the diagonal matrix with the entries of $e_k$ on its diagonal. So, the $E_k$ matrices capture the structure of $D$ while respecting the basis directions. The coefficient matrices $E_k$ are important here since they generate a representation of the Clifford algebra $\mathfrak{Cl}(\mathbb{R}^{|V|}, q)$, with $q$ giving the quadratic form on $\mathbb{R}^{|V|}$ as before. Specifically, these matrices satisfy: $E_i E_j + E_j E_i = -2q(e_i, e_j)\mathbf{Id}_{|V|}$ where $q(e_i, e_j)$ denotes the quadratic form evaluated on basis elements and $\mathbf{Id}$ is the identity matrix.

**Remark 2.** *This discrete formulation of the Dirac operator on graphs parallels the continuous case in differential geometry. Understanding this link helps interpret how $D$ acts on a function $f$ over the vertices of $G$ and informs practical design choices: **(i)** For real-valued functions, $(Df)(v)$ can be a weighted average of values at neighboring vertices. **(ii)** For vector-valued functions, $D$ acts on each component independently, encoding more complex relationships. **(iii)** For complex-valued functions, $D$ incorporates phase information. **(iv)** For spinor-valued functions, $D$ acts on $\mathcal{S}(G)$, the space of $\mathbb{C}^{2^{\lfloor |V|/2 \rfloor}}$-valued functions, with $D_{vw}$ as complex matrices. Specifically, for $\psi \in \mathcal{S}(G)$, the Dirac operator acts as: $(D\psi)(v) = \sum_{w \in V} D_{vw}\psi(w)$ where $D_{vw}$ are $2^{\lfloor |V|/2 \rfloor} \times 2^{\lfloor |V|/2 \rfloor}$ complex matrices. This case has roots in quantum mechanics, and provide useful heuristics.*

**Spinors and Fock space.** For the complex Clifford Algebra, there exists an irreducible representation $\phi : \mathfrak{Cl}_{|V|}(\mathbb{C}, q) \to \text{End}(\mathbb{S})$, where $\text{End}(\mathbb{S})$ denotes the space of linear endomorphisms of $\mathbb{S}$, and $\mathbb{S}$ is a complex vector space of dimension $2^{\lfloor |V|/2 \rfloor}$, called the **Spinor** space (Lounesto, 2001). Note that $\text{End}(W)$ and $\text{Hom}(W, W)$ are essentially the same object. The Spinor space is relevant because of the following result: the Spinor space $\mathbb{S}$ can be identified with the exterior algebra $\wedge(\mathbb{C}^{\lfloor |V|/2 \rfloor})$, which is *isomorphic* to the **Fock** space $\mathbb{F} = \bigoplus_{k=0}^{\lfloor |V|/2 \rfloor} \wedge^k(\mathbb{C}^{\lfloor |V|/2 \rfloor})$. This isomorphism allows us to work with the Fock space representation instead of the complete Clifford algebra. *Why is this useful?* Recall that the Dirac operator uses only the

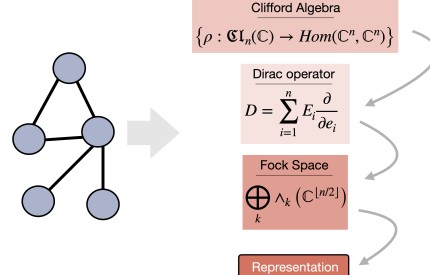

Figure 3: A schematic to go from graph to Fock space representations.

basis elements $E$'s of the Clifford algebra. These basis elements act on spinors, which can be identified with elements of the Fock space. The action of $E_k$ on the Fock space can be decomposed into so-called creation and annihilation operators: $E_k \simeq a_k + a_k^*$ where $a_k$ is the annihilation operator and $a_k^*$ is the creation operator. By using only the basis elements, we can significantly simplify our computations while retaining the essential structure of the Clifford algebra: we can work directly with creation and annihilation operators.

**Remark 3.** *As an alternative, it is possible to represent and identify elements of the Clifford algebra with structures in infinite-dimensional Hilbert spaces where each vertex of $G$ will be treated as an element in a one-particle Hilbert space. Our approach above is more direct, naturally accommodates the finite-dimensional nature of our graph while still providing a rich algebraic structure. Importantly, sensible approximations will be available.*

## 2.3 TRANSLATING THEORY TO PRACTICE: INSTANTIATING A GRAPH REPRESENTATION

The Fock space formulation provides a framework for representing multi-particle systems, analogous to encoding features from a graph. However, implementing the full structure, especially in high dimensions, can be challenging. Vector Symbolic Architectures (VSA), as explored in recent works (Alam et al., 2023; Ganesan et al., 2021), offer a practical approximation of Fock spaces with computational efficiency. In VSA, the binding operation (circular convolution) approximates the creation/annihilation operators, while the superposition operation (vector addition) resembles the direct sum in Fock spaces. Although the VSA$\leftrightarrow$ quantum mechanics connection is not new (Wolff et al., 2018), in this context it helps use the power of Fock spaces while offering computational efficiency.

**Representing nodes, sums, and products.** In our implementation, we assign a high-dimensional vector to each concept (node, edge, and so on). These vectors play a role analogous to the basis elements in the expansion of the Dirac operator. While ideally, these vectors would be orthogonal,

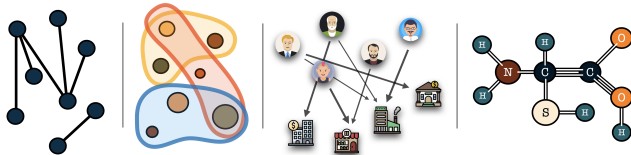

Figure 4: Graphs, Hypergraphs, Attributed graphs, Proteins. All these types of graphs can be efficiently encoded using FoGE.

similar to the properties of basis elements in a Fock space, we simply approximate this by sampling from a normal distribution $\mathcal{N}(\mathbf{0}, 1/d)$. This leads to nearly orthogonal vectors, with the maximum absolute cosine similarity between any two vectors typically below $0.1$ (Blum et al., 2020).

For operations analogous to those in Fock space, we use dimensionality-preserving operations instead of tensor products, which significantly increase dimensionality (Wolff et al., 2018). This simplifies implementation, as all resultant embeddings maintain the same dimensionality regardless of the encoding method. We define the sum ($\oplus$) as element-wise addition and the product ($\otimes$) as circular convolution; an operation analogous to the creation and annihilation operators in Fock space. This can be implemented as element-wise multiplication of the vectors' Fourier representations followed by an inverse Fourier transformation. Notice that, as $d$ grows, these operations asymptotically approach the algebraic properties of Fock space (i.e., the probability that we violate the algebraic properties of Clifford algebra goes to zero) while its complexity is $\mathcal{O}(d \log d)$. This scheme also allows us to define the inverse vector, i.e., for any vector $b$, we have a vector $a$ such that the identity $a \otimes b = \mathbf{1}$ holds. It is known that other properties like commutation relations, super-position and self-commutation are also mostly satisfied in VSA. Note that our experiments are not tied to this specific implementation, and improved choices can be dropped in.

**Dealing with infinitely many concepts.** In some datasets in our experiments, each vertex comes with a text description. Defining one vector per word or sentence at random is not ideal anymore. To avoid this problem, we use a text encoder. Models like CLIP (Radford et al., 2021), BERT (Devlin et al., 2019), RoBERTa (Liu et al., 2019), and others are effective at mapping whole text passages to vectors in a way that the information is preserved while similar sentences are mapped to similar areas in space. So, we initialize our construction by defining the vectors as such an encoding. In this way, (a) we can create infinitely many vectors, and (b) similar vectors represent similar concepts. When the dimensionality permits, we keep the default sampling approach, and note the use of text-encoders explicitly in the experiments.

**Other Works using Vector Symbolic Architectures.** There is a growing body of results in the literature using Vector Symbolic Architecture (VSA) (Schlegel et al., 2022) albeit for other problems. The idea has its roots in *symbolic AI*, where VSA sought to benefit from the high dimensional representations in addition to well-defined *logical* rules for combining these symbols/vectors in some manner. Many works using VSA describe the construction mechanistically, deriving specific ways to generate the underlying symbols as well as implementing the "merge" operations. The possibility of using Fock space for symbolic manipulation (Wolff et al., 2018) has been identified by others with example results reported regarding its utility in trajectory analysis. Vector symbolic representations have also been recently used for computational efficiency considerations related to self-attention calculation in HRRFormers (Plate, 1995; Alam et al., 2023).

## 3 FOCK GRAPH ENCODER (FOGE)

Based on the concepts from Section §2, we use a parameter-free scheme (denoted FoGE) to obtain rich graph embeddings. Our approach is general and can handle a large spectrum of different graph types, and its extension to novel graph-types is straightforward. Diverse graph types such as hypergraphs, attributed graphs, as well as proteins (Fig. 4) can all be modeled easily providing an alternative or a good initialization for more intensive trainable models. This approach translates the abstract concepts of Fock spaces into a practical and efficient method for graph representation, where graph features obtained by the encoding are analogous to multi-particle states in a Fock space.

For a graph $G = (V, E)$ we have a vector $\mathbf{p}_i$, using $i$ to index the nodes. We also use an extra vector $\mathbf{s}$ for the graph's size, a practical design choice we will explain shortly. Then, with these $n + 1$ vectors, we obtain a lossless Fock-space based representation $\mathbf{g}$ as:

$$\mathbf{g} = \left(\mathbf{s} \otimes \mathbf{p}_n\right) \oplus \bigoplus_{(i,j) \in E} \left(\mathbf{p}_i \otimes \mathbf{p}_j\right) \tag{1}$$

Our formulation follows from §2. Each edge's endpoints are fused together using $\otimes$ and then we aggregate all edges together using $\oplus$. Finally, the graph's size is also added using the special vector $\mathbf{s}$.

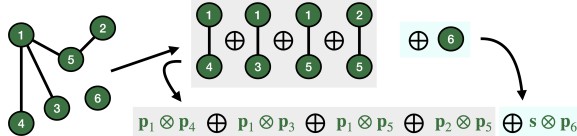

**Lossless representation.** The above representation is lossless. Assuming we use (1) to get a graph's embedding $\mathbf{g}$. Then, simply by evaluating the expression $\mathbf{p}_j^T(\mathbf{p}_i^{-1} \otimes \mathbf{g})$, we can determine whether the edge $(i, j)$ exists in the edge set of that particular graph. In this way, we can recover, one by one, all edges of the graph and correctly reconstruct it, if desired. It is instructive to check the importance of $\mathbf{s}$. By evaluating the expression $\mathbf{p}_i^T(\mathbf{s}^{-1} \otimes \mathbf{g})$, $\forall i$, we can first obtain the size of the graph. This can inform the edge retrieval above because an expression of the form $\mathbf{p}_{n+x}^T(\mathbf{p}_i^{-1} \otimes \mathbf{g})$ could, in practice, produce a number close to 1, although there is no such edge. By first obtaining the size of the graph, we have a "safeguard" against such phantom edges beyond the real vertex-set.

**Vertex attributes.** Consider a graph $G = (V, E, Attr)$, where the set $Attr$ (with $|Attr| = |V|$) consists of attributes, one for each vertex. There is no restriction on the type of attributes: it can denote numerical values or text or any other concept. Let $\mathbf{a}_i$ be the vector associated with the attribute of vertex $i \in V$ (using an appropriate text-encoder if needed). Then, we can augment (1) to absorb the extra information in the following way:

$$\mathbf{g} = \left(\mathbf{s} \otimes \mathbf{p}_n\right) \oplus \bigoplus_{(i,j) \in E} \left(\mathbf{p}_i \otimes \mathbf{p}_j\right) \oplus \bigoplus_{i \in V} \left(\mathbf{p}_i \otimes \mathbf{a}_i\right) \tag{2}$$

The graph is again, fully reconstructable. We have also encoded each vertex's attribute (which can be recovered by the expression $\mathbf{a}_j^T(\mathbf{p}_i^{-1} \otimes \mathbf{g})$). We should think of proteins as a graph with vertex attributes where each vertex is a specific amino acid (possibly with 3-D coordinates).

**Hypergraphs (Theory versus Practice).** Hypergraphs are generalizations of graphs: each edge is connected to an arbitrary number of vertices, instead of just 2 (Fig. 4). In theory, we can easily augment (1) so that we can handle hypergraphs as follows:

$$\mathbf{g} = \left(\mathbf{s} \otimes \mathbf{p}_n\right) \oplus \bigoplus_{(k_1, \cdots k_m) \in E} \bigotimes_{i=1}^{m} \mathbf{p}_{k_i} \tag{3}$$

In practice, aggregating many multiple vectors together may be unstable. This is true for our particular design choices for calculations (e.g., circular convolution), so we use an alternative approach. We can start by observing that each edge can be interpreted as a unique cluster of vertices, so we simply assign a unique vector $\mathbf{e}_i$, $i \in \left[|E|\right]$ to each edge in the hypergraph. This modification allows us to encode the hypergraph similar to how a graph is encoded as a dictionary, in the following way:

$$\mathbf{g} = \left(\mathbf{s} \otimes \mathbf{p}_n\right) \oplus \left(\bigoplus_{i=1}^{|E|} \left(\mathbf{e}_i \otimes \bigoplus_{j \in E_i} \mathbf{p}_j\right)\right) \tag{4}$$

### 3.1 FOCK SPACE-BASED GROUNDING OF LLMS (FOGE-LLM)

Recent works showed that (a) textualizing a graph and pre-appending it to a question results in better-than-random responses from the LLM (although far from perfect), and (b) using a specialized graph encoder such as a GNN or a graph transformer and training along with a frozen LLM results in a big improvement in performance, resulting essentially in LLMs that can understand, to some

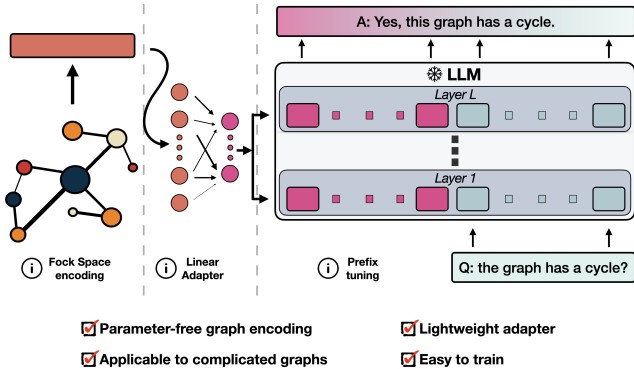

Figure 5: FoGE-LLM overview. Using a parameter-free graph encoder we get graph embeddings for a range of different graphs. Then, we use linear adapters with a frozen LLM for *prefix tuning*.

extent, graphical structures. One takeaway is that we can bypass the most tedious stage of designing application-specific graph encoders. Instead, we can use a parameter-free method for a wide range of graph types, as we described above. Thus, the only trainable parts of the pipeline are simple linear adapters that convert the raw graph encodings to a format "understandable" by an LLM. Our FoGE-LLM is shown in Fig. 5. After getting the graph encodings, we train one/more linear adapters and append the transformed encodings to the question's embeddings fed to the LLM.

**Summary and Takeaway.** We highlight some qualitative advantages. *First*, our graph encoding is parameter-free and efficient. The complexity of aggregation is $\mathcal{O}(d \log d)$ ($d$ is vectors' dimension) and the number of aggregation operations is linear (in graph size). *Second*, our encoder is not restricted to specific graph types: it works easily for simple graphs, for proteins and for hypergraphs just via small modifications. In contrast, GraphToken (Perozzi et al., 2024) uses a specific GNN whose output size is dependent on the underlying task whereas GraphLLM (Chai et al., 2023) uses a transformer model together with a GNN (also specific to the underlying task). These properties simplify our training and eliminates any tunable components. *Third*, our open-source code offers a scalable way to train FoGE-LLM even on consumer GPUs, by using FSDP (Zhao et al., 2023). As a reference, GraphToken (Perozzi et al., 2024) is trained on TPUs (code unavailable) whereas GraphLLM (Chai et al., 2023) has a large memory/compute footprint (trained on A100 80GB).

## 4 EXPERIMENTAL RESULTS

We examine our Fock-space based encoding in two separate settings: (a) as a stand-alone input of a simple model, and (b) as an extra prefix in a frozen LLM (FoGE-LLM), for graph prompting.

**Datasets and Models.** We performed experiments on multiple graph reasoning datasets: from simple graph-understanding tasks to hypergraphs and proteins and aim to cover different aspects of graph understanding/reasoning. Specifically, we consider the 6 following datasets/dataset collections: **(i) GraphQA** (Fatemi et al., 2024) **(ii) GraphReasoning** (Chai et al., 2023) **(iii) HyperGraphQA (iv) PPI** (Hamilton et al., 2017) **(v) OBNB** (Liu & Krishnan, 2024) **(vi) SabDab** (Dunbar et al., 2013). More details about the datasets can be found in the appendix. Exploring

Table 1: Results on two real-world protein datasets from OBNB. Our method appears to be the stronger unsupervised scheme to obtain node embeddings, especially for the DisGeNet task. Its performance is comparable to trainable, graph-specific models (GCN and GAT). More details on all baselines are in (Liu & Krishnan, 2024).

| Model | BioGRID | | HumanNet | |
| --- | --- | --- | --- | --- |
| | DisGeNet | GOBP | DisGeNet | GOBP |
| LabelProp | 0.931 | 1.885 | 3.059 | 3.806 |
| Adj + LR | 0.743 | 2.528 | 3.053 | 3.964 |
| Node2Vec + LR | 0.836 | 2.571 | 2.433 | **4.036** |
| LapEigMap + LR | 0.864 | 2.149 | 2.301 | 3.778 |
| **FoGE** | **1.062** | 2.433 | **3.254** | 3.916 |
| GCN (Bruna et al., 2014) | 1.012 | **2.572** | 3.116 | 3.812 |
| GAT (Liu & Zhou, 2020) | **1.063** | 2.562 | 3.065 | 3.963 |

various graph reasoning datasets allows us to analyze the performance and generalization capabilities of our proposed model across different graph structures and domains. From traditional graph-based

question-answering tasks to more complex hypergraph understanding and biological network analysis. By tackling these varied datasets, we aim to gain a comprehensive understanding of the capabilities and limitations of our approaches in graph reasoning tasks. Additionally, the inclusion of real-world datasets such as PPI, BioGRID, and HumanNet underscores the practical relevance of our research, with potential applications in biological research, network analysis, and beyond.

We use the Llama2 (7B) model (Touvron et al., 2023) as the frozen LLM, and we employ only extra linear adapters for the graph embeddings we obtain using our formulation. We adjust vector dimensionality from $512$ to $2048$ and use just a *single adapter* for the entire model or *one adapter per layer* in FoGE-LLM.

## 4.1 Proof of Principle Evaluations for Graph Understanding

Table 2: Using a small neural network with a single layer on the obtained graph representations allows us to perform almost perfectly in tasks such as *number of nodes* and *number of edges* in a graph, for both synthetic and real data.

|  | GraphQA | | | HyperGraphQA | | Jaffe | |
|  | num nodes | num edges | has cycle | num nodes | num edges | num acids | num links |
|---|---|---|---|---|---|---|---|
| MSE/Acc | 0.67 | 0.03 | 98.7% | 1.12 | 0.63 | 2.95 | 11.9 |
| Model size | 32K | 8K | 16K | 32K | 4K | 32K | 16K |

**Setup and Results.** While our key goal is graph-prompting, we first perform multiple preliminary checks of the effectiveness of our graph encoding. We conduct three different types of experiments.

*First*, we evaluate whether our graph embeddings are informative (i.e., they preserve the graph's structure), by using a small, 1-hidden-layer FFN for basic graph-understanding tasks, e.g., *number of nodes* and *edges*. We use 3 different classes of graphs (simple graphs, hypergraphs, and proteins) and the results show that our representations are rich and informative (Table 2) and only few parameters suffice to achieve almost-perfect performance on such tasks.

*Second*, we examine whether our graph encodings preserve important biological markers of the data. To test this, we use a small dataset of about 900 proteins (SabDab) which are accompanied by affinity data that corresponds to each protein's clade. Briefly, clades are protein superfamilies, based on common ancestry (more information can be found in the appendix). In theory, proteins from the same clade are *more similar* than across clades, so we examine whether this is also preserved in our obtained embeddings. Although the dataset has only few samples and some of the clades are scarcely populated, we can observe that there is a clear separation between the most populated clades in the embeddings space (Fig. 8).

*Third*, we examine if the same encoding practice can generate rich node-level encodings, by encoding for each node, the subgraph that is generated by itself and its neighbors. We examine the performance in **nineteen** real protein datasets (PPI (Hamilton et al., 2017) and OBNB (Liu & Krishnan, 2024)) in Tables 1, 3, and 9. We see that our approach is, in all datasets, among the best unsupervised approaches, and is also competitive (if not better) than specialized supervised approaches that leverage trainable, graph-specific models such as GCN (Bruna et al., 2014) and GAT (Liu & Zhou, 2020). Specifically, we achieve state-of-the-art performance in PPI while we are the best-performing method (among both unsupervised and supervised) in seven out of the eighteen datasets of OBNB.

These results provide encouraging evidence that (a) our approach gives "rich" graph embeddings for a range of different graph types and styles, and (b) our graph embeddings can be used as an extra, grounding input to a powerful LLM without the need to design/train a specialized model, e.g., GNN (Scarselli et al., 2009; Wu et al., 2022) or a Graph Transformer (Dwivedi & Bresson, 2020).

Table 3: Micro F1-score on PPI. Our approach is better than the best unsupervised approaches and better/comparable to the supervised approaches.

| | Model | F1 |
|---|---|---|
| | Random | 39.2 |
| | Node2Vec (Yun et al., 2022) | 40.9 |
| | Raw features (Yun et al., 2022) | 42.2 |
| Unsupervised | GraphSAGE-min (Hamilton et al., 2017) | 46.5 |
| | GraphSAGE-max (Hamilton et al., 2017) | 50.2 |
| | DGI (Veličković et al., 2019) | 63.8 |
| | GRACE (Zhu et al., 2020) | 66.2 |
| | **FoGE** | **99.2** |
| Supervised | GraphSAGE-min (Hamilton et al., 2017) | 50.0 |
| | GraphSAGE-max (Hamilton et al., 2017) | 61.2 |
| | LGCN (Gao et al., 2018) | 77.2 |
| | GAT (Liu & Zhou, 2020) | 97.3 |
| | GCNII (Chen et al., 2020) | **99.5** |

## 4.2 GROUNDING LLMs WITH GRAPH PROMPTING

**Graph Understanding.** In our first experiment, we examine whether an LLM can answer questions about a graph's structure, such as the number of nodes, the presence of cycles, and so on. We use GraphToken and conduct a suite of six different experiments. Although our method's encodings are not specific to each underlying task, it performs competitively with specialized models, as shown in Table 4. Even when GraphToken uses different embeddings for each node (*node degree*) or edge (*edge existence*), our model still achieves comparable results using a single embedding for the entire graph, except for node degree prediction, where GraphToken's node-specific embeddings offer an advantage.

Table 4: GraphToken vs FoGE-LLM on GraphQA. Column *1* stands for a single embedding for the entire graph; $\mathcal{O}(n)$ stands for a single embedding per node. In all 6 tasks, although we use a parameter-free, predetermined graph encoding, we see a performance similar/better relative to a trainable graph encoder linked with a larger LLM (PaLM-2). For reference, we also include the best performance with any ICL-based technique (Fatemi et al., 2024; Perozzi et al., 2024).

|  | ICL | GraphToken | | FoGE-LLM |
|---|---|---|---|---|
| Tokens | $\mathcal{O}(n^2)$ | 1 | $\mathcal{O}(n)$ | 1 |
| num of nodes | 26.9% | **99.6**% | - | 97.2% |
| num of edges | 12.8% | 42.6% | - | **45.1**% |
| cycle existence | 83.2% | 95.6% | - | **97.9**% |
| num of triangles | 16.2% | 34.8% | - | **37.7**% |
| node degree | 28.0% | - | 96.2% | 62.7% |
| edge existence | 54.4% | - | 73.8% | **74.3**% |

**Advanced Graph Reasoning** Going beyond "simple" graph understanding tasks, we also examine our performance on more complicated graph-reasoning tasks, using a recent dataset (Chai et al., 2023). GraphToken is not applicable here since each node is accompanied by a textual description which cannot be handle by that model. So, our main baseline is GraphLLM, which uses a transformer combined with a GNN to merge the graphical/textual information into one or more embedding vectors. Similar to GraphToken (Perozzi et al., 2024), GraphLLM (Chai et al., 2023) also utilizes a different approach for each task, using multiple graph embeddings for each task. In contrast, we achieve comparable performance using a *single graph embedding*, showcasing the versatility/richness of the graph embeddings. Further, we see that using a pretrained text encoder such as RoBERTa (Liu et al., 2019) to generate the vectors is reasonable, and results in a similar performance.

Table 5: GraphLLM vs FoGE-LLM. Although we are using the same, predetermined graph embedding for each task, we enjoy a performance similar to GraphLLM which leverages 5 graph embeddings, specific to the task at hand. The *vectors* stands for the two approaches we follow in generating them: (a) randomly generated (almost) orthogonal vectors (ignoring the node's text description), and (b) using RoBERTa (Liu et al., 2019) and utilizing all vertices' information.

|  | GraphLLM | **FoGE-LLM** | |
|---|---|---|---|
| model size | 100M | **25M** | |
| question specific output | Yes | **No** | |
| graph embeddings | 5 | **1** | |
| vectors | - | random | RoBERTa |
| substructure count | 99.9% | 97.3% | 95.6% |
| max triplet sum | 95.7% | 94.6% | 94.7% |
| shortest path | 97.2% | 95.7% | 95.8% |
| bipartite match | 99.8% | 98.1% | 97.3% |

This is a strong improvement over traditional symbolic methods, by allowing a large set of "symbols"/vectors. Dealing with proteins is similar to advanced graph reasoning, since both datasets are graphs with additional node information. In Table 6, we show the accuracy of FoGE-LLM for three protein-related tasks on Jaffe. Although the size of the proteinic graphs is more than $10\times$ larger compared to the ones in GraphQA and GraphReasoning, our model is able, up to some extent, to understand the provided protein, as a whole (*number of amino acids* and *number of links*) as well as at an individual-node level for the task *type of amino acid* (where we prompt the model to determine the type of a specific vertex in the protein).

**Hypergraphs.** Existing works focus on specific forms of graphs and rarely applicable (or easily modifiable) to different graph types. One common family of graphs in applications is hypergraphs. Here each edge is a subset of the nodes, of arbitrary size (Fig. 4). Our formulation can handle such a generalization of the typical graphs with only minor modifications to the encoding formulation (Eq. 4). Here, we show that our design can indeed answer questions about such complicated structures, using our encodings as an extra prefix

Table 6: FoGE-LLM performance against ICL techniques for hypergraphs and proteins.

|  |  | Zero-Shot | Few-Shot | **FoGE-LLM** |
|---|---|---|---|---|
| HyperQA | num of nodes | 04.5% | 16.8% | **85.0**% |
|  | num of edges | 03.9% | 27.0% | **95.4**% |
|  | node degree | 02.1% | 10.1% | **53.9**% |
|  | edge existence | 65.9% | 79.4% | **87.9**% |
| Jaffe | num of amino-acids | 03.9% | 17.1% | **99.3**% |
|  | num of links | 03.8% | 06.1% | **13.2**% |
|  | amino-acid type | 01.4% | 12.3% | **37.7**% |

(graph prompting). Using the proposed dataset (HyperGraphQA), we assess the performance of

FoGE-LLM on four common tasks. Since GraphToken as well as GraphLLM cannot handle such data, we compare our model's performance against two of the most common prompt-engineering methods: 1. zero-shot, where the model is given the graph in text form along with the corresponding question, and 2. few-shot, where the model is given pairs of textualized graphs with the corresponding question/answer pair and it is asked to produce the answer to a new combination of graph/question. The results are presented in Table 6. Interestingly, even though hypergraphs have a much more complicated structure than "simple" graphs, our model achieves a performance very close to basic graph understanding (Table 4), or even better at some tasks.

## 5 RELATED WORK

**Geometric Algebra in Machine Learning.** There is growing interest in application of geometric algebra in machine learning, particularly for developing neural networks that maintain geometric properties. While these ideas have been leveraged in the context of equivariance/symmetry transformations in deep learning (Cohen et al., 2019; Bronstein et al., 2021; Banerjee et al., 2022; Zhu et al., 2018; Finzi et al., 2020), the theory is finding interesting uses in recent works. For example, (Zhdanov et al., 2024) recently proposed Clifford Neural Layers to model dynamical systems in fields like fluid dynamics and (Ruhe et al., 2023b) described Geometric Clifford Algebra Networks (GCANs), specifically designed to respect symmetry group transformations. Beyond classical machine learning, geometric algebra finds more direct applications in quantum computing as well: (Trindade et al., 2023) leveraged the isomorphism between Pauli matrices and Clifford Algebra to represent multidimensional data, to define specialized transforms for machine learning tasks.

**Graphs & LLMs.** The body of work describing ways to infuse extra, graphical information into a frozen LLM is sizable and growing. As discussed earlier, initial approaches focused on converting the underlying graph into natural language form, such as "node 1 is connected to node 3, node 5 is connected to node 4, ..." (Wang et al., 2023; Guo et al., 2023; Fatemi et al., 2024). These works while far from perfect showed viability: that a frozen LLM has the capability to reason about the given graph and answer graph-related questions, such as "is there a cycle in the graph?". Practical difficulties involving the format of graph serialization is an important factor in the performance and the results tend to be only moderately better than random. The perspective taken in (Perozzi et al., 2024; Chai et al., 2023) was fresh and led to an alternative approach: infusing the graph information directly at the embedding level, by encoding the graph using a model such as a Graph Neural Network (GNN) (Scarselli et al., 2009; Wu et al., 2022; Perozzi et al., 2024) or a Graph Transformer (Dwivedi & Bresson, 2020; Chai et al., 2023). These works significantly improved the state of the art, showing that carefully crafted graph embeddings are key to a successful grounding of an LLM.

## 6 CONCLUSIONS

We have described a novel strategy to encode a graph into a vector form for direct downstream use or to augment prompts fed to LLMs. Our approach, grounded in Clifford algebra and Fock space operations, is rigorous and offers numerous advantages in practice demonstrated via experiments. We can obtain encodings of arbitrary graphs instantly, with no trainable parameters that nicely encapsulates the important information content in the underlying graph. Using these encodings, we introduced FoGE-LLM, a way to fuse the graph information for graph-prompting with a pre-trained, frozen LLM, allowing it to "understand" and reason about graphs. Our model, accompanied with a simple-to-train open-source codebase, performs favorably relative to highly specialized models while at the same time handling classes of graphs where other alternatives fall short or need adjustments.

**Limitations.** A key strength of our method is its generality; it is a parameter-free way to obtain rich graph embeddings. However, if our method fails to produce informative-enough graph embeddings for graphs in a specific application, the parameter-free nature offers us very few knobs to turn to improve the performance. The fix, we believe, is to build representation learners on top of the embeddings derived by our model which we conjecture is happening to some extent in FoGE-LLM anyway. Additionally, as noted before, when we need to deal with a potentially infinitely large set of vectors, generating them at random is infeasible. While, for our experiments, RoBERTa appears to be a sensible option, the efficacy may not translate to other new datasets.

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

## A  DATASET DETAILS

In our experiments, we used the following datasets:

1. **GraphQA** (Fatemi et al., 2024): It includes 6 different graph-understanding tasks (*number of nodes*, *number of edges*, *cycle existence*, *number of triangles*, *node degree*, and *edge existence*) on 7 different graph structures (Erdos-Renyi (Erdös & Rényi, 1959), Scale-Free, Barabasi-Albert (Albert & Barabási, 2002), Stochastic Block Model, Star, Path and Complete).

2. **GraphReasoning** (Chai et al., 2023): Recently introduced in (Chai et al., 2023) to better assess the model's graph understanding ability, it consists of 4 more advanced graph-understanding tasks (*substructure count*, *maximum triplet sum*, *shortest path*, and *bipartite graph matching*). Each graph node is accompanied by extra information in the form of a text description, making this dataset a suitable testbed for our RoBERTa-based vector encoding.

3. **HyperGraphQA**: We extend GraphQA to Hypergraphs. Specifically, we consider 4 different graph-understanding tasks (*number of nodes*, *number of edges*, *node degree*, and *edge existence*) on 2 different hypergraph structures (Erdos-Renyi (Erdös & Rényi, 1959), and Chung-Lu (Chung & Lu, 2002)). The training dataset consists of only 2000 instances, making it hard for large models to avoid overfitting.

4. **Jaffe** (Jaffe et al., 2022): Jaffe is a recent dataset consisting of approximately 1.6 million natively paired human antibody sequences from healthy donors. To our knowledge, this represents by far the largest publicly available dataset of its kind.

5. **PPI** (Hamilton et al., 2017): PPI consists of 24 proteins collected from human tissue, with each node associated with 121 binary labels. Compiled from experimental techniques like yeast two-hybrid screening and mass spectrometry, as well as computational predictions, such a dataset provides critical insights into the functional organization of the proteome. By understanding how proteins interact, scientists can uncover the molecular underpinnings of cellular processes and develop targeted therapeutic strategies.

6. **OBNB** (Liu & Krishnan, 2024): OBNB (Open Biomedical Network Benchmark) is a collection of 15 datasets (including well-known datasets such as BioGRID (Stark et al., 2006) and HumanNet (Hwang et al., 2019)). Each dataset's sample consists of a gene accompanied by 3 vectors (named *DISEASES*, *DisGeNET*, *GOBP*) of node-level binary labels.

7. **SabDab** (Dunbar et al., 2013): SabDab (Structural Antibody Database) is a collection of 919 publicly available, annotated antibody structures (proteins). Each structure is accompanied by multiple annotations, such as the heavy and light chain pairing.

## B  FOGE-LLM: TRAINING DETAILS

We train the LLM-based construction with a batch size of 16 and a learning rate of $1e$-3. The model required less than 10 epochs to convergence, in contrast to other works that require more training time due to the ellaborate graph encoders (e.g., Chai et al. (2023)). Our implementation is based on Pytorch Lightning Falcon & The PyTorch Lightning team (2019), which allows us to split and train the model on multiple GPUs using FSDP. This implementation allows the user to train this, or any similar, model to conventional GPUs with less memory while, at the same time, speed up the process by preloading all the obtained lightweight graph embeddings to the GPUs. The *merging* of the graph embedding with the LLM is based on the idea of prefix tuning Li & Liang (2021), i.e., pre-append the embedding to the input text embeddings and, in our case, this is happening with the use of a linear adapter. We experimented both with a single linear adapter on the input layer, as well as a linear adapter per layer and the difference was only marginal in the final results.

## C  FOGE-LLM: INFERENCE DETAILS

Besides the low training time, FoGE-LLM enjoys an extremely low inference time, due to two reasons. First, we always "reserve" only a single token for the provided graph. In contrast, zero/few-shot

approaches that textualize the graph require a large number of tokens, prohibitively large as the graph grows. This leads to an explosion of the inference time, due to the transformer's quadratic dependency on the number of input tokens. Second, FoGE-LLM employs one or more linear adapters and does not require any specialized architectures, like existing solutions Chai et al. (2023); Perozzi et al. (2024). This, as we observed in our experiments, impacts the inference time, casting FoGE-LLM one of the fastest graph-augmented Language Models. In Table 7 we present the average inference time required for each approach.

| Model | Inference time (s) $\downarrow$ |
|---|---|
| zero-shot | 0.175 ($\pm$0.05) |
| few-shot | 0.541 ($\pm$0.10) |
| GraphLLM Chai et al. (2023) | 0.052 ($\pm$0.01) |
| **FoGE-LLM** | 0.031 ($\pm$0.01) |

Table 7: Average inference time for each approach on Llama-7B. FoGE-LLM is significantly lower than zero/few shot approaches since the number of input tokens does not grow with the graph size, while it enjoys a $40\%$ improvement over GraphLLM dues to its simpler encoder/adapter.

## D ICL PROMPTING FOR HYPERGRAPHS

In Table 6 we demonstrate FoGE's superiority over In-Context Learning approaches, like zero-shot and few-shot prompting. Here we explain how we created the textual descriptions of the hypergraphs, that were used in both zero- and few-shot prompting. Following similar works for graph textualization Perozzi et al. (2024); Fatemi et al. (2024), we first assign a number to each node and then, in a new line, we explain which nodes are part of each hyperedge. An example can be seen below.

> G describes a hypergraph among 0, 1, 2, 3, 4, 5, 6, 7, and 8.
> In this hypergraph:
> Hyperedge 1 connects nodes 2, 3, 6.
> Hyperedge 2 connects nodes 1, 4, 5, 7.
> Hyperedge 3 connects nodes 1, 2.
> Hyperedge 4 connects nodes 3, 5, 7, 8.

After the hypergraph textualization, the question follows in the case of zero-shot, while both the question and the answer follow in the case of few-shot.

## E LOSSLESS REPRESENTATIONS

One advantage of the obtained embeddings is that fact that the underlying structures are recoverable. This allows us to obtain unbiased vector estimates of complicated structures, such as graphs with multiple edge and node attributes. Here, we show how this property manifests in our specific formulation as well as more generally for pairs of key-item.

### E.1 CAPACITY

One of the typical ways to examine the performance of such a construction is by assuming a vector $\mathbf{u}$ as being the bundling of multiple binded pairs, as in the following equation

$$\mathbf{u} = \bigoplus_{i=1}^{n} \mathbf{k}_i \otimes \mathbf{v}_i \tag{5}$$

and then examine how accurately we can recover each vector $\mathbf{v}_i$, given the corresponding $\mathbf{k}_i$. In theory, the vector $\mathbf{v}_i$ can be easily recovered using the operation:

$$\tilde{\mathbf{v}}_i = \mathbf{k}_i^{-1} \otimes \mathbf{u} \tag{6}$$

In Fig. 6 we examine the cosine similarity of the obtain vector $\tilde{\mathbf{v}}_i$ with the correct one ($\mathbf{v}_i$) as well as with all the rest ($\{\mathbf{v}_j\}_{j \neq i}$). We observe that the results follow closely the theoretical results above, with a perfect separation of up to 100 pairs, and a small overlap for $200 - 300$ pairs of vectors.

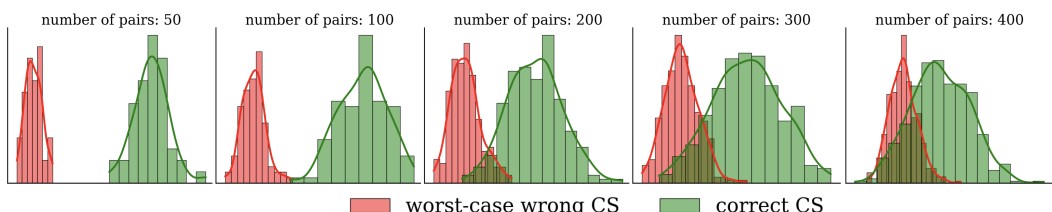

Figure 6: Given a vector $\mathbb{R}^{4096} \ni \mathbf{u} = \oplus_{i=1}^{n} \mathbf{k}_i \otimes \mathbf{v}_i$, how correctly we can recover all pairs of keys-values back, as the number of pairs ($n$) grows. *Worst-case wrong CS* corresponds to the maximum cosine similarity of the recovered value vector with all value vectors but the correct one, and *correct CS* corresponds to the cosine similarity with the correct value vector.

## E.2 GRAPH RECONSTRUCTION

In our specific application, we deal with graphs and, as we analyzed in §3, the graph representations we obtain are, in theory, lossless, i.e., we can recover back the original graph from the vector representation using the inverse vectors. Here, we examine whether this claim holds in practice too. In Fig. 7 you can observe the strength of each edge after reconstruction, for 3 different vector dimensionalities. We can observe that, even for a moderately large dimension, there is a clear separation between the true edge set and the rest of the edges.

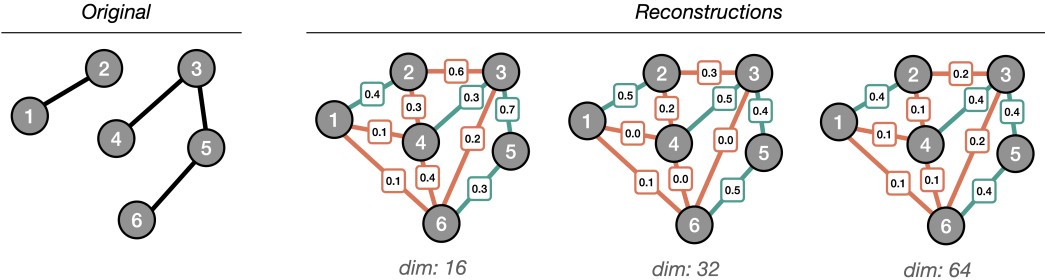

Figure 7: Lossless representations: even for small vector dimension, we can obtain back the true edge set. The numbers show the cosine similarity of the obtained vector with the true edge vector, and it can be used to estimate the true edge set.

## F    PRESERVATION OF CLADE INFORMATION ON SABDAB

Given that the SabDab proteins (Dunbar et al., 2013) are annotated with the heavy/light chain pairing, we can extract the clades and visualize their embeddings with respect to that information. As a brief reminder, the clades correspond to superfamilies of proteins that share a common ancestor (Han et al., 2007). To extract the clades we used the V gene heavy chain and chose seven families. It is well known from biology that antibodies that belong to the same clade are *more similar* than antibodies across different clades, so, here, we examine if this real-world, biological property is reflected on our embeddings. Specifically, after obtaining each protein's embedding using FoGE (in an unsupervised fashion without using the clade annotations), we apply a T-SNE transformation on the high-dimensional vectors so that we are able to plot them, with a significant amount of noise, in just two dimensions. Although we reduce the dimensionality significantly, and, even worse, we deal with a extremely small dataset of just 919 proteins (Table 8), in Fig. 8 we can observe that the proteins of each clade cluster together. This is a different, qualitative indicator, which shows that FoGE is able to preserve all the information that is encapsulated in the inputted structures.

Table 8: Distribution of samples across the different clades. In total there are 919 samples, with clades 1, 3, 4 being the most frequent.

| Clade | 1 | 2 | 3 | 4 | 5 | 6 | 7 | *Total* |
|-------|-----|-----|-----|-----|-----|-----|-----|---------|
| **Count** | 325 | 28 | 414 | 101 | 25 | 3 | 23 | 919 |

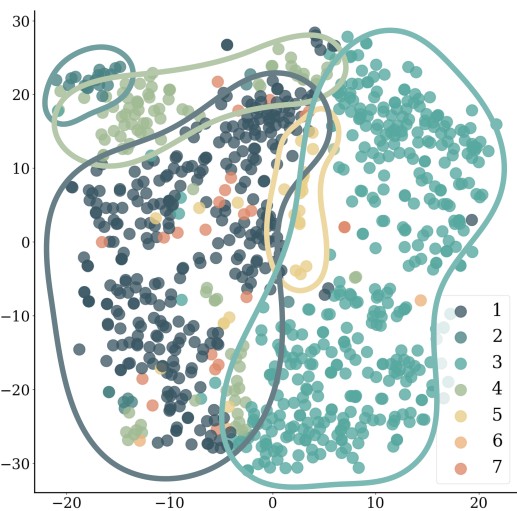

Figure 8: T-SNE plot of the SabDab embeddings. Although the dataset is very small, each one of the populated clades occupies a different region and, interestingly, clades 1 and 7 are very similar, just like in real life. The T-SNE plot was robust to different choices of hyperparameters, with no significant differences beyond simple translations of the space.

## G    ADDITIONAL RESULTS ON OBNB

OBNB (which stands for Open Biomedical Network Benchmark) is a collection of multiple, real-world protein datasets, where each node (or amino-acid) of each protein is accompanied by multiple binary labels. A detailed analysis of the datasets and their labels can be found in (Liu & Krishnan, 2024) and the corresponding repository. In Table 9 we present the results on all 18 reported datasets of OBNB. FoGE is one of the best-performing methods across all benchmarks, showcasing once more the capabilities of our obtained embeddings.

## H    IMPACT OF VECTOR DIMENSION

One of few the hyperparameters of FoGE is the dimensionality of the vectors (i.e. graph embeddings). Using GraphQA, we perform an ablation study on the impact of the dimension on the final accuracy of the model (Fig. 9). Relative accuracy is calculated as the actual accuracy for each dimensionality, divided by the best one, for each task respectively, and it allows us to compare different tasks with completely different best performances (Table 4).

From this study, a few important remarks surface that we observe to hold true for the other datasets too. First of all, a larger dimensionality does not always "translate" to better results. We observe that for some tasks (*cycle existence*), we achieve the optimal performance with a dimension significantly lower than the maximum we consider (2048), matching essentially GraphToken's performance with less than 20K trainable parameters, while in some cases there is a small drop as we go from 1024 to 2048. Finally, as with most of the tunable hyperparameters in machine learning models, there is no predetermined best strategy for choosing the dimensionality. For instance, when we consider *cycle existence* or *the number of triangles* we can have a highly performing model with a dimensionality of less than 128, while for tasks such as *edge and node count* the performance drops significantly as we reduce the dimensionality.

Table 9: FoGE vs multiple unsupervised and supervised methods. After obtaining our embeddings, we use a Random Forest to predict the corresponding node's label. The evaluation is based on the APOP metric (Liu & Krishnan, 2024) and we can observe that FoGE is always comparable to the best methods, while in almost half of the cases it is the best one.

| Network | Model | DISEASES | DisGeNET | GOBP |
|---|---|---|---|---|
| BioGRID | LabelProp | 1.210 | 0.931 | 1.858 |
| | LogReg | 1.556 | 1.026 | 2.571 |
| | GCN+BoT | 1.511 | 1.014 | 2.442 |
| | SAGE+BoT | 1.486 | 1.031 | 2.402 |
| | GIN+BoT | 1.410 | 1.007 | 2.386 |
| | GAT+BoT | **1.609** | 1.037 | **2.624** |
| | GatedGCN+BoT | 1.547 | 1.038 | 2.517 |
| | **FoGE** | 1.599 | **1.062** | 2.433 |
| HumanNet | LabelProp | 3.728 | 3.098 | 3.806 |
| | LogReg | 3.812 | 3.158 | **4.053** |
| | GCN+BoT | 3.552 | 3.053 | 3.921 |
| | SAGE+BoT | 3.401 | 3.052 | 3.816 |
| | GIN+BoT | 3.513 | 3.054 | 3.861 |
| | GAT+BoT | 3.761 | 3.100 | 3.809 |
| | GatedGCN+BoT | 3.677 | 3.086 | 3.889 |
| | **FoGE** | **3.853** | **3.254** | 3.916 |
| COMPPIHumanInt | LabelProp | 1.352 | 1.106 | 2.076 |
| | LogReg | 1.644 | 1.240 | **2.806** |
| | GCN+BoT | 1.648 | 1.211 | 2.685 |
| | SAGE+BoT | **1.694** | 1.210 | 2.629 |
| | GIN+BoT | 1.608 | 1.219 | 2.611 |
| | GAT+BoT | 1.665 | 1.230 | 2.755 |
| | GatedGCN+BoT | 1.672 | 1.218 | 2.735 |
| | **FoGE** | 1.660 | **1.241** | 2.586 |
| BioPlex | LabelProp | 0.964 | 0.939 | 1.714 |
| | LogReg | **1.358** | **0.939** | 2.587 |
| | GCN+BoT | 1.324 | 0.911 | 2.553 |
| | SAGE+BoT | 1.246 | 0.865 | 2.513 |
| | GIN+BoT | 1.349 | 0.868 | 2.504 |
| | GAT+BoT | 1.355 | 0.873 | 2.548 |
| | GatedGCN+BoT | 1.301 | 0.859 | 2.590 |
| | **FoGE** | 1.273 | 0.879 | **2.599** |
| HuRI | LabelProp | 0.545 | 0.598 | 1.086 |
| | LogReg | 0.650 | 0.656 | 1.084 |
| | GCN+BoT | 0.634 | 0.693 | 1.129 |
| | SAGE+BoT | 0.593 | 0.679 | 1.190 |
| | GIN+BoT | 0.583 | 0.702 | 1.143 |
| | GAT+BoT | 0.667 | 0.687 | 1.174 |
| | GatedGCN+BoT | 0.596 | 0.695 | **1.195** |
| | **FoGE** | **0.684** | **0.729** | 1.070 |
| OmniPath | LabelProp | 1.358 | 0.897 | 1.593 |
| | LogReg | 1.542 | **1.093** | **2.125** |
| | GCN+BoT | **1.577** | 1.068 | 2.071 |
| | SAGE+BoT | 1.478 | 1.062 | 1.986 |
| | GIN+BoT | 1.452 | 1.073 | 1.993 |
| | GAT+BoT | 1.552 | 1.048 | 2.068 |
| | GatedGCN+BoT | 1.516 | 1.049 | 2.071 |
| | **FoGE** | 1.511 | 1.085 | 2.102 |

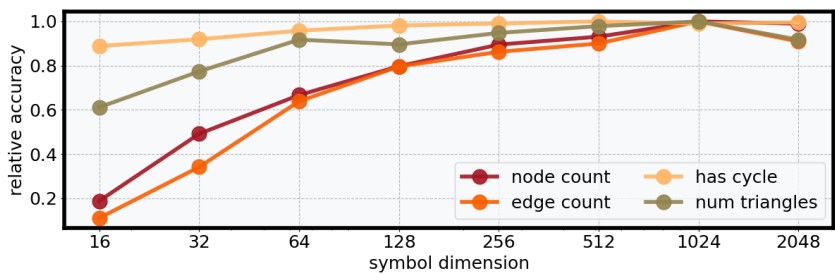

Figure 9: Accuracy versus vectors dimensionality. Although there is a positive trend between the two quantities, the dependency on the dimension is not equally strong or always positive in all tasks.

