# OpenReview forum: "FoGE: Fock Space inspired encoding for graph prompting"
_ICLR.cc/2025/Conference — Submitted to ICLR 2025_

### Official Review · Reviewer_nWt3 · 2024-10-23

**Soundness:** 3
**Presentation:** 2
**Contribution:** 2
**Rating:** 6
**Confidence:** 3

**Summary:**

This paper proposes FoGE, which is a parameter-free Fock-space-based method to generate graph representations. The method can encode arbitrary graphs into embeddings and train a linear adapter that aligns these embeddings with a frozen LLM via prefix-tuning. It can handle various graph types, including simple graphs, hypergraphs, and attributed graphs, and the experiments show that this method can achieve competitive performance with baseline models.

**Strengths:**

- This paper proposes a theoretical lossless Fock-space-based method to generate graph representation and can handle graphs with attributes. The use of parameter-free encoding simplifies the design compared to other GNNs or graph transformers.
- This method generalizes across various types of graphs and the experiments validate its performance on many tasks. It generates graph embedding that is not specific to each task and performs better than other specialized models in some cases.

**Weaknesses:**

- It is not clear whether the LLM contributes to understanding graph structures or merely functions as a text generation module. The interaction between the LLM and graph embeddings should be further explored. For example, it would be insightful to see how the graph embeddings perform if they were used to train a simple MLP or other lightweight models for downstream tasks. This would help clarify whether the LLM adds value beyond generating natural language outputs.
- In Table 5, the proposed method underperforms relative to GraphLLM across several tasks, which brings concerns about whether FoGE can consistently achieve better performance in advanced graph reasoning tasks.
- The experimental settings are not thoroughly discussed, and details about the training process and hyperparameters are missing. Providing more transparent implementation details would improve the reproducibility of this work.

**Questions:**

- Could you explain how to obtain the size vector $s$ in the encoding process?
- Table 4 highlights the performance on hypergraphs compared to zero-shot and few-shot scenarios. Could you provide more details on how to generate the corresponding prompts?

---

> ### Author Response · Authors · 2024-11-21
> **Use of simple MLP on the graph embeddings**
>
> We thank the reviewer for their time and effort. We appreciate their positive comments and we clarify their questions below.
>
> **Use of simple MLP on the graph embeddings**
>
> This question is about disentangling the contributions of our graph encoder from the LLM's capabilities. Our response below should clarify this doubt completely because much of this is already included in the paper but perhaps not identified as such.
>
> Our experiments show that FoGE provides rich, informative graph embeddings even without an LLM. The encoder's effectiveness is checked through multiple independent analyses:
> our graph embeddings capture complex structural patterns even with simple models (Section 4.1). More compelling is their ability to preserve sophisticated biological relationships - the embeddings naturally cluster proteins by their evolutionary clades without any explicit evolution-related signals (see Appendix E). This shows that our encoder captures meaningful high-order interactions. The encoder's strength is further validated through extensive protein network experiments on PPI and OBNB datasets (Tables 1, 3, 8), where it outperforms both unsupervised and supervised baselines. Notably, using just our embeddings with small MLPs achieves impressive performance on classic graph tasks (please see Table 2).
> Subsequently, when combined with an LLM, we can do more. This integration is powerful because our encoder preserves complete graph information while mapping it into a form amenable to the LLM's reasoning capabilities. The LLM is helping generate natural language output, but it is also doing more — it is performing structural reasoning using the preserved graph patterns, much like how LLMs process other non-textual modalities when given appropriate representations.

---

> ### Author Response · Authors · 2024-11-21
> **Comparison with GraphLLM**
>
> **Comparison with GraphLLM**
>
> The performance difference requires careful contextualization. While GraphLLM shows marginally better numbers, it achieves this through a significantly more complex architecture: (A) a 100M parameter graph encoder versus our 25M total parameters, (B) 5 task-specific graph embeddings versus our single encoding, and (C) custom output heads for each task. Despite these advantages, our parameter-free encoding with simple linear adapters comes within 2% of GraphLLM's performance.
>
> But more importantly, GraphLLM's task-specific design limits its applicability: it cannot handle hypergraphs or protein structures without some architectural modifications. In contrast, our single framework demonstrates strong performance across diverse graph types, from simple graphs to complex biological networks, achieving excellent results on protein tasks.

---

> ### Author Response · Authors · 2024-11-21
> **More transparent implementation**
>
> **More transparent implementation**
>
> We take reproducibility seriously and should clarify that our code will be publicly available. We appreciate the reviewer’s constructive comment and have added an extra paragraph to the Appendix to explain the hyperparameters used. However, note that our implementation is quite simple, and the hyperparameters correspond only to the vectors’ dimension (e.g., 1024), the learning rate, and the batch size.

---

> ### Author Response · Authors · 2024-11-21
> **How the size vector is obtained**
>
> **How the size vector is obtained**
>
> Our first step is to calculate the total number of vectors needed. For instance, for our simple example below Eq. (1), the total number of vectors would be 6 (one for each node — denoted $p_i$) + 1 (the size indicator —denoted $s$). After we have estimated the total number of vectors (7 in our example), we create 7 high-dimensional vectors orthogonal to each other (or almost orthogonal if 7 is larger than the chosen dimensionality). If we denote as $v$ the collection of these vectors, then $p_i=v_i,\ i=1,\cdots 6$, and $s=v_7$. In general, the whole set of vectors is chosen to be orthogonal to each other, or almost orthogonal if the number of vectors surpasses the dimensionality, as we detail in Section 2.3, without any specific treatment to any vectors/concepts.

---

> ### Author Response · Authors · 2024-11-21
> **Zero-shot details**
>
> **Zero-shot details**
>
> We thank the reviewer for the comment. We followed the prompting style of the baselines, and we have updated the Appendix to include more information as well as examples of how the prompts look like.

---

> ### Author Response · Authors · 2024-11-25
>
> Dear Reviewer nWt3,
>
> we sincerely value your positive feedback and the constructive comments for our paper. As the discussion period approaches its conclusion, we want to confirm that our responses meet your expectations and effectively address your concerns.

---

### Official Review · Reviewer_ydJ1 · 2024-10-30

**Soundness:** 2
**Presentation:** 2
**Contribution:** 2
**Rating:** 5
**Confidence:** 1

**Summary:**

The paper presents a parameter-free graph encoder using Fock space representations to improve LLMs in answering graph-related questions. This simple approach provides rich encodings for diverse graphs, enabling effective prefix-tuned prompts for pre-trained LLMs, simplifying and generalizing existing methods.

**Strengths:**

Integrating graph encoders with LLMs to address graph-related tasks is a convincing approach.

**Weaknesses:**

1. **Unclear motivation**: The motivation of this paper is not clear. Many previous works leverage graph encoders to learn graph representations for integration with LLMs [1,2], and the overall framework appears similar.
2. **Potential confusion with graph prompting methods**: In this work, the authors use graphs to prompt LLMs, which could be confused with existing graph prompting methods [3,4,5]. The authors should discuss the differences between these approaches to help readers better understand the purpose of this work.
3. **Impact of graph encoder choice**: Will the choice of graph encoder affect the performance of the proposed method? Some analytical experiments may provide a better understanding of the impact of the graph encoder choice.


[1] Chen, Runjin, et al. "LLaGA: Large Language and Graph Assistant." Forty-first International Conference on Machine Learning.\
[2] Tang, Jiabin, et al. "Graphgpt: Graph instruction tuning for large language models." Proceedings of the 47th International ACM SIGIR Conference on Research and Development in Information Retrieval. 2024.\
[3] Liu, Zemin, et al. "Graphprompt: Unifying pre-training and downstream tasks for graph neural networks." Proceedings of the ACM Web Conference 2023. 2023.\
[4] Sun, Mingchen, et al. "Gppt: Graph pre-training and prompt tuning to generalize graph neural networks." Proceedings of the 28th ACM SIGKDD Conference on Knowledge Discovery and Data Mining. 2022.\
[5] Yu, Xingtong, et al. "Few-shot learning on graphs: from meta-learning to pre-training and prompting." arXiv preprint arXiv:2402.01440 (2024).

**Questions:**

See weaknesses.

---

> ### Author Response · Authors · 2024-11-21
> **Unclear motivation**
>
> We thank the reviewer for their time and effort. We thank them also for the additional works mentioned which we have already included in the updated manuscript. Below we analyze the points that the raised and we remain at their disposal for any further questions.
>
> **Unclear motivation**: Many previous works leverage graph encoders to learn graph representations
>
> Yes, as noted in our paper, several proposals for integrating graphical information into an LLM have been proposed. We include comparisons with several prominent baselines. Integrating graphs into an LLM is not the core contribution of the paper. This paper deals with how to do so effectively and efficiently. We should that a single linear layer adapter is sufficient provided the encodings from our core FoGE module. This is very different from alternatives proposed in the literature.
>
> Below we briefly review how each work differs from FoGE-LLM (sections 3.1 and 4.2 of our work):
>
> 1. LLaGA converts the graph (or some part of the graph) into a textual representation, which existing results already show can be expensive since the text grows with the size of the graph. Our graph representation is of a fixed size. Additionally, LLaGA requires a text encoder and an MLP adapter on top to align the graph information. We only need a linear layer adapter.
> 2. GraphGPT uses a GNN backbone, to convert the graph to multiple embedding vectors and a text encoder to encode the textual information of the nodes to multiple embedding vectors too. In contrast, we use a single, parameter-free encoder for both the graph structure and the additional textual information: the full structure is converted to a single embedding vector.

---

> ### Author Response · Authors · 2024-11-21
> **Confusion with graph prompting methods**
>
> **Confusion with graph prompting methods**
>
> Our approach is a graph prompting method: it uses graph structure to condition an LLM's behavior through input augmentation. Like other prompting approaches, we modify the LLM's input representations to incorporate graph information, but via a different mechanism. The key difference is how this conditioning is achieved: through parameter-free Fock space encodings rather than trained GNN representations.
>
> Our work differs fundamentally from [3,4,5] in both philosophy and technical approach. While these works focus on training procedures for GNN-based graph encoders, we introduce a novel parameter-free encoding. We can briefly summarize the key technical distinctions:
> 1. *Parameter-free versus Trainable*: Our encoder requires no training, unlike GNN-based approaches in [3,4,5].
> 2. *Foundations*: Our encoder is derived from significantly different mathematical principles.
> 3. *Versatility*: FoGE naturally handles multiple graph types without architectural changes.
>
> While these works (references to these results are now in the paper) have advanced graph prompting through innovative training strategies, our contribution is orthogonal in that we give a theoretically grounded, efficient encoding mechanism. Our method can potentially benefit from their training insights, but addresses a different question: how to represent graphs for LLM consumption.

---

> ### Author Response · Authors · 2024-11-21
> **Impact of graph encoder choice**
>
> **Impact of graph encoder choice**: Will the choice of graph encoder affect the performance of the proposed method?
>
> We are not sure we understand the question. A new graph encoder is the central technical contribution (section 2). It is analyzed in detail in section 3. In section 4 we show its benefits as a standalone model (section 4.1) and as part of an LLM (section 4.2). We will be glad if the reviewer can clarify the question.

---

> ### Author Response · Authors · 2024-11-25
>
> Dear Reviewer ydJ1,
>
> we sincerely value your positive feedback and the constructive comments for our paper. As the discussion period approaches its conclusion, we want to confirm that our responses meet your expectations and effectively address your concerns.

---

> > ### Comment · Reviewer_ydJ1 · 2024-11-29
> >
> > Thank you for your response. I will provide a final score after reading the responses from other reviewers.

---

> > > ### Author Response · Authors · 2024-11-30
> > >
> > > Thank you for the response. Please let us know if we have addressed all of your concerns and if we can answer anything else.

---

### Official Review · Reviewer_DL79 · 2024-11-01

**Soundness:** 3
**Presentation:** 2
**Contribution:** 3
**Rating:** 5
**Confidence:** 3

**Summary:**

This paper proposed a novel approach to obtain powerful and model-agnostic graph representations that can be used as prompts to augment LLM’s capabilities of answering graph-related questions. By leveraging Fock spaces, a concept from mathematical physics, they achieved almost lossless task-agnostic graph embeddings capturing the diverse graph structure and information. A lightweight linear adapter is then adopted to map the rich Fock-space inspired embeddings into an LLM’s embedding space, making prefix-tuning effective for various graph-related tasks.

**Strengths:**

1. By leveraging Fock-space inspired graph encoding, this approach achieves nearly lossless parameter-free graph embeddings, containing rich graph structure information.
2. They show that a simple linear layer can map the task-agnostic graph embeddings into the LLM embedding space, offering a low-computational approach to align graph representations with pre-trained language models.
3. They conducted experiments to validate the informativeness of Fock-space inspired graph embeddings on basic graph-understanding tasks using a small neural network. Experiments also show that their embedding method performs the best in unsupervised approaches and is competitive with specialized supervised methods.
4. This graph encoding method can adapt to different graph types, node-level embeddings and hypergraphs, making it suitable for a wide range of applications.
5. They demonstrated that by using fock-space graph embedding, the LLM can do better at graph understanding and reasoning.

**Weaknesses:**

1. Although the method is task-agnostic, it might underperform on specialized tasks without additional tuning.
2. Although the method can achieve nearly lossless parameter-free graph embeddings, there’s no study whether it can effectively capture high-order interactions between nodes, which is important in complex graph reasoning.

**Questions:**

1. Can you explain how the node representations $p_i$ and extra vector size vector $s$ are obtained? Will these representations affect the Fock-space inspired embeddings?
2. How much performance gap is this task-agnostic graph encoding method with specially tuned ones?
3. Can Fock-space inspired graph encoding understand high-order interactions between nodes?

---

> ### Author Response · Authors · 2024-11-21
> **Comparison with specialized methods**
>
> We thank the reviewer for appreciating our work and for the multiple strength points they mention. Below we analyze the points that they raised and we remain at their disposal for any further questions.
>
> **Might under-perform on specialized tasks without additional tuning**
>
> Yes, this is indeed noted in the limitations paragraph of our conclusion. However, extensive experimentation (Tables 1, 2, 3, 8, and Figure 8) on multiple real and synthetic datasets did not show any signs of weakness compared to methods like GNN and GAT. We believe that often the effort in designing and debugging a specialized architecture for a task is not negligible, and in most cases, some additional tuning on a specialized task is an acceptable (or even preferable) option.
>
> **How much is the performance gap relative to specially tuned ones?**
>
> As noted above, we did not observe a performance gap in any of our experiments with FoGE and FoGE-LLM. We see this in Tables 1, 2, 3, 4, 5, 6, 8, where we compared FoGE with many of the typical graph encoder models.

---

> ### Author Response · Authors · 2024-11-21
> **Can it capture high-order interactions?**
>
> **Can it capture high-order interactions?**
>
> Thanks for bringing up this point. We can analyze this both conceptually and from a practical standpoint.
>
> Theoretically, the Fock space formalism inherently represents multi-particle states. This is directly analogous to higher-order graph interactions.
>
> In practice, we are indeed able to capture the high-order interaction. We see this when FoGE is able to create protein graph embeddings that cluster nicely based on the clade information, although *no* relative signal is used during the encoding (Appendix E). This experiment shows that FoGE is able to “understand” the underlying graphs and the node interactions, by clustering the proteins based on their superfamilies (clades), acknowledged to be a difficult problem (e.g., SCOP: a Structural Classification of Proteins database, The I-TASSER Suite: protein structure and function prediction). Additionally, the protein classification tasks (Tables 2, 3, 8) as well as the core graph tasks we consider in Table 5 (e.g., shortest path, substructure count) require of the model to understand the node interactions, since such tasks can not be answered correctly by a shallow model that does not capture these high-order interactions. If there are any additional experiments that the reviewer suggests we include to emphasize this point more strongly, we are happy to include it in the paper.

---

> ### Author Response · Authors · 2024-11-21
> **How node representations are obtained?**
>
> **How node representations are obtained?**
>
> In Section 2.3,  we describe the specifics our our practical instantiation. Briefly, when the number of vectors allows (i.e., the number of the total vectors is lower than their dimension), we choose the vectors to be orthogonal. Otherwise, we sample them from the normal distribution, resulting in almost orthogonal vectors (e.g., https://www.cs.princeton.edu/courses/archive/fall14/cos521/lecnotes/lec11.pdf). In our experiments, we see that random permutations of the vectors or the choice of a different basis do not affect the quality of the encoding and the results of the downstream tasks were almost identical on different runs. Specifically for the size vector s, we do not make any special design choices besides asking that it should be (almost) orthogonal relative to the rest of the vectors.
>
> *A simple example*: For instance, for our simple example below Eq. (1), the total number of vectors would be 6 (one for each node — denoted $p_i$) + 1 (the size indicator —denoted $s$). After we have estimated the total number of vectors (7 in our example), we create 7 high dimensional vectors orthogonal to each other (or almost orthogonal if 7 is larger than the chosen dimensionality). If we denote as $v$ the collection of these vectors, then $p_i=v_i,\ i=1,\cdots 6$ and $s=v_7$. In general, the whole set of vectors is chosen to be orthogonal to each other, or almost orthogonal if the number of vectors surpasses the dimensionality, as we detail in Section 2.3, without any specific treatment to any vectors/concepts.

---

> ### Author Response · Authors · 2024-11-25
>
> Dear Reviewer DL79,
>
> we sincerely value your positive feedback and the constructive comments for our paper. As the discussion period approaches its conclusion, we want to confirm that our responses meet your expectations and effectively address your concerns.

---

### Official Review · Reviewer_TDtw · 2024-11-02

**Soundness:** 2
**Presentation:** 2
**Contribution:** 2
**Rating:** 5
**Confidence:** 4

**Summary:**

Inspired by Fock space, the paper proposes a training-free graph encoder approach to align graph embeddings with the LLM's embedding space. Specifically, the paper uses a parameter-free scheme to obtain graph embeddings and then trains a linear layer to align these embeddings with the LLM's embedding space, enabling the model to handle graph tasks effectively with minimal adjustments to the architecture.

**Strengths:**

it is a novel idea to use fock space inspired method to obtain graph embedding.

**Weaknesses:**

- Lack of efficiency experiments. I agree with the authors that the graph encoding is parameter-free and efficient. However, the complexity is dominated by the LLM rather than the GNN, even though the LLM is frozen. When using large LLMs such as LLaMA-7B, the overall training time will not differ significantly whether the graph encoder requires training or not. Therefore, my concern is that the training-free graph encoder does not offer a noticeable efficiency improvement in terms of real training/application.

- Justification of the use of LLMs. Why LLM should be used in graph reasoning task (without text attribute). It make senses to introduce LLM to help with textual graph tasks, because such tasks need the text reasoning ability, world knowledge from LLM. However, for traditional graph tasks such substructure count, shortest path etc, I didn;t see the necessity of using LLMs.

- Comparison with RAG. In the introduction, the authors mention RAG and compare it with prefix tuning. However, I don’t see the relevance of this comparison. Prefix tuning is designed to adapt a model’s attention to new contexts with minimal parameter updates, whereas RAG combines retrieval mechanisms with generation to enrich the model’s knowledge base dynamically. The distinction between the two methods is significant, and it would help if the authors clarified why RAG was introduced here or provided a more targeted comparison relevant to graph encoding.

- Imprecise Statements. The statement, "RAG-based approaches for graphs primarily involve converting graphs to text, while prefix tuning with graphs uses modules to extract richer, task-relevant structures, requiring larger sample sizes and higher compute power," is unclear. Could you add references to RAG-based approaches where "graphs primarily involve converting graphs to text"? For example, in [1], retrieval is performed over graphs instead of text, and in [2], text is organized into graphs for retrieval.

References:

[1] G-Retriever: Retrieval-Augmented Generation for Textual Graph Understanding and Question Answering, NeurIPS 2024, https://arxiv.org/pdf/2402.07630

[2] From Local to Global: A Graph RAG Approach to Query-Focused Summarization, https://arxiv.org/pdf/2404.16130

**Questions:**

Do you need to train separate adapters for each dataset, or is there a unified adapter for all datasets?

---

> ### Author Response · Authors · 2024-11-21
> **Justification of the use of LLMs**
>
> We thank the review for their time and effort. Below we clarify the main points identified in the review which should resolve the key concerns.
>
> **Justification of the use of LLMs**
>
> The rationale behind why LLMs are needed or will be effective for any datatype that is non-text-based touches on a shift in how modern LLMs are viewed. Results in the last two years give strong evidence supporting LLMs as general-purpose reasoning engines (not just text processors). This perspective change is prominent along two distinct paths: for use in language+X settings or even for completely new modalities unrelated to text. Importantly, in both cases, the direct connection of modality X to language can be tenuous or abstract.
>
> These two paths are popularized via (a) multi-modal language models and (b) language models for X (for some X).  Already, a sizable body of work has developed: Vision LM (https://arxiv.org/abs/2305.06500), Video LMs (https://arxiv.org/abs/2311.10122), Graph LMs (https://arxiv.org/pdf/2402.08170), LLMs for Tabular Data (e.g., https://arxiv.org/pdf/2410.04739), LLMs for Time series data (https://arxiv.org/abs/2310.01728) as well as the influential ESM (https://www.biorxiv.org/content/10.1101/2022.07.20.500902v1) and ProtGPT (https://pubmed.ncbi.nlm.nih.gov/35896542/) for protein structure design. ESM-2 and ESMFold directly process raw amino acid sequences (which are pure chemical sequences with no linguistic properties) to predict protein structure and function. We hope that our description clarifies that repurposing LLMs, either via piggybacking on the LLM capabilities (but for a new modality X) or using these powerful models for general-purpose reasoning is being intensively studied.
>
> A quick clarification on graph tasks: indeed, we do not need to invoke an LLM to ask substructure count and shortest path questions: this is not the highlight of this work. Instead, these examples showcase the functionality of the model — in other words, if the encoding is sensible, we should be able to get sensible answers. We still find this result/capability interesting and we also describe how the construction alone can yield substantial improvements over traditional approaches (33% improvement on PPI, state-of-the-art results on 7 out of 18 OBNB datasets) on real-world datasets. The key point is to notice the distinction between specialized models that excel at narrow tasks versus a generalist approach.

---

> ### Author Response · Authors · 2024-11-21
> **Lack of efficiency experiments**
>
> **Lack of efficiency experiments**
>
> There are several points we want to make which should clarify this doubt completely:
>
> 1. For a specific task, a user has a choice between a highly specialized graph network tailored to the task versus a LLM-based alternative (described here and in several other works). So, based on the reasons in the answer above, if a practitioner decides to use LLMs for their graph tasks (versatility, reasoning capabilities, or other benefits), the relevant question is: what is the most effective way to encode the graphs for LLM consumption? Here, our parameter-free approach offers clear advantages over alternatives like GraphToken and GraphLLM. Some advantages include handling multiple graph types without architecture changes, mathematically sound lossless representations, simple adapters, minimal optimization or hyperparameter overhead, and comparable/better performance.
> 2. We now address the concern that the training time will not differ, whether the graph encoder needs training or not. We should highlight three key points. First, the fact that FoGE is parameter-free allows us to encode all graphs offline, before even using the LLM. As the reviewer agrees, FoGE is efficient, lightweight, and parallelizable, so we can obtain quickly the graph embeddings. Second, the fact that the graph embeddings have already been obtained implies that the data loading and transfer time from the GPU and back is minimized, or even eliminated if the dataset is not extremely large. On the contrary, it is impossible to preload all graphs to the GPU memory beforehand, which means that GNN-based models need to also deal with this bottleneck and pay a cost. Third, our trainable parameters are only linear layers and not a specialized architecture. Yes, the parameter-count is dominated by the LLM, but this does not mean that there will be no difference in the training time and the time to convergence between FoGE-LLM and a Graph Language Model that employs a much more elaborate construction for graph encoding. For example, GraphLLM was trained for 20 epochs, while FoGE-LLM needed less than 10. Additionally, the time difference between preloading the data to the GPU and transferring them for each iteration is more than 11%, and this improvement will be higher in more resource-restricted regimes (which our implementation fully supports).
> 3. The comment about "real training/application" suggests that our paper is focused only on simple graph tasks (which was only intended as an interesting first experiment). In practice, we see real strengths on multiple real-life datasets (see Tables 1, 3, 8) where other methods can struggle. Additionally, we note that this application domain frequently requires quick adaptation to new graph structures/properties. From a practical perspective, deployment efficiency is often an important criterion for the adoption of a method. Our formulation's simplicity (and broad penetration/software maturity of publicly available LLM implementations) means faster deployment cycles, easier debugging, and more straightforward integration into pipelines.

---

> > ### Comment · Reviewer_TDtw · 2024-11-25
> > **Response to Lack of efficiency experiments**
> >
> > Thank you for addressing my questions. Here are my responses, point by point.
> >
> > 1. To encode the graphs for LLM consumption in FoGE, we still need to fine-tune the linear projection layer, right?
> > 2. Could you please provide the end-to-end training time or training throughput for an efficiency comparison? From the statement, "GraphLLM was trained for 20 epochs, while FoGE-LLM needed less than 10," we cannot conclude that FoGE-LLM is more efficient since we don't know the training time per epoch. Regarding the speedup in the data preloading part, could you explain how much time it actually takes? I am not sure it would be a bottleneck compared to the training.
> > 3. Agreed.

---

> ### Author Response · Authors · 2024-11-21
> **Comparison with RAG/Imprecise Statements**
>
> **Comparison with RAG/Imprecise Statements**
>
> We apologize for the confusion. Our comparison was specifically focused on how additional information is incorporated into the LLM's input - in RAG's generation phase, this involves concatenating retrieved information with the prompt, similar to how prefix-tuning augments the input. We acknowledge that the terminology was not accurately chosen, as In-Context Learning (ICL) is a more suitable term for describing the graph-to-text conversion approaches we were discussing. This is a localized change limited to 2-3 lines, we have already updated the manuscript to avoid any further confusion. We would appreciate any more feedback if there is still room for confusion.
>
> As per the reviewer’s request, we list some results that involve “converting graphs to text”:  https://arxiv.org/pdf/2405.20139 , https://arxiv.org/pdf/2307.07697 , https://arxiv.org/pdf/2306.04136 , https://arxiv.org/pdf/2002.08909 , https://arxiv.org/pdf/2310.04560 . Even the Reference [2] mentioned in the review, textualizes the graphs before feeding them to the LLM (see section 2.5) and does not use any graph encoders.

---

> > ### Comment · Reviewer_TDtw · 2024-11-25
> > **Response to Comparison with RAG/Imprecise Statements**
> >
> > Thank you for the revisions. The revised manuscript looks good to me.
> >
> > As for the RAG-related work, it seems we are not on the same page. It appears I am referring to the retrieval process, where retrieval is based on graphs, whereas the authors are discussing the generation process, where the retrieved subgraph is textualized and provided as a prompt to the LLM. For example, the mentioned references, https://arxiv.org/pdf/2405.20139 and https://arxiv.org/pdf/2307.07697, are works that implement retrieval on graphs and then textualize the retrieved subgraph. However, https://arxiv.org/pdf/2310.04560 does not seem to be related to Graph RAG.

---

> ### Author Response · Authors · 2024-11-21
> **Separate adapters**
>
> **Separate adapters**
>
> For an exact/fair comparison with the baselines, we train one adapter per dataset similar to the baselines. Training one adapter for multiple datasets, i.e., creating a foundational graph+text model, would require significantly larger and diverse training datasets (like those used in existing foundational multi-modal models) and although possible in principle, would involve a very extensive data curation exercise and matching financial resources.

---

> > ### Comment · Reviewer_TDtw · 2024-11-25
> > **Response to Separate adapters**
> >
> > Thank you for addressing my question.
> >
> > Overall, I am willing to raise my score to 5.

---

> ### Comment · Reviewer_TDtw · 2024-11-25
> **Response to Justification of the use of LLMs**
>
> Thank you for the justification. I agree with the authors. It would be great to see a discussion of this justification for the use of LLMs in the paper.

---

> ### Author Response · Authors · 2024-11-26
>
> We thank the reviewer for their response. We are glad that we were able to clarify many of the questions. We hope that the following answer will clarify any remaining doubts.
>
> ### **Justification of the use of LLMs**
>
> We are have expanded our abstract and introduction and incorporated our discussion about LLMs usage. Thank you for the recommendation!
>
> ### **Efficiency experiments**
>
> Thank you for the comments.
>
> 1. **Does the linear adapter need training?** That is correct, the linear projection layer has to be trained before FoGE-LLM can be used. This is not different from other proposals that combine a graph encoder with an LLM, although in our case the trainable part is much smaller and simpler.
> 2. Regarding the runtime of FoGE-LLM:
>     1. **FoGE-LLM verus GraphLLM (training time):** GraphLLM has a slower convergence behavior and it requires more epochs. Here, we provide more quantitative details about each model’s performance (the raw numbers relate to the “substructure count” task while the percentages correspond to averages across all datasets). Using the same batch size for a fair comparison, FoGE-LLM is about 30% faster for each epoch. Using A100 GPUs, FoGE-LLM requires 100 seconds while GraphLLM requires about 140 seconds per epoch. Additionally, FoGE-LLM converges after 10 epochs while GraphLLM requires at least 20. This means that the improvement is about 1800 seconds, or about 60%. Notice also that, since GraphLLM uses a transformer-based text encoder, this difference is even larger for graphs with rich textual information (e.g., publications/citations networks).
>     2. **FoGE-LLM versus GraphLLM (inference time):** Due to our model’s simplicity, we observe that the inference time of FoGE-LLM is lower (something that is also reflected in the lower per-epoch time). Speficially, on average, FoGE-LLM requires 0.03 seconds per sample, while GraphLLM requires 0.05 (about 40% improvement).
>     3. **Data preloading versus transferring:** Besides a direct comparison with GraphLLM, we should note a key feature of our approach is the offline nature of our encoder, where sizable gains can be harvested. Given that we the encoding process is lightweight, parameter-free, and parallelizable, we can first encode all graphs and preload the corresponding graph embeddings to the GPU. To highlight the improvement, we trained two instances of FoGE-LLM, one with all the data preloaded to the GPU and another one in which we transferred the batch data back and forth from the GPU on each forward pass, a training procedure that resembles most of the other Graph Language Models. Using GraphQA as the training dataset, each epoch takes about 50 and 60 seconds respectively to run (a 16% improvement). Using other datasets, we observed a similar pattern, with the minimum improvement observed at about 11%.
>
>     To summarize, even if we put aside other properties of our formulation, from a purely wall-clock time perspective, FoGE-LLM significantly improves the training time of Graph Language Models compared to other GNN+LLM approaches, both due to its simple adapters as well as its lightweight, parameter-free, and parallelizable graph encoder. Additionally, the inference time is orders of magnitude lower than graph textualization approaches whose input grows with the size of the underlying graph. We appreciate the comment and we have added these extra information to the appendix (sections B, C) to highlight this. While the improvement is in the scale of minutes due to the size of the datasets, the relative improvement shows that FoGE-LLM can be used a general-purpose Graph Language Model with significantly less computational effort.
>
>
> ### **Comparison with RAG/Imprecise Statements**
>
> We now understand that the reviewer meant retrieval, and why our earlier response may have been unclear. Yes we believe that the reviewer is correct — our work addresses a different part of the graph to LLM pipeline. While results like https://arxiv.org/pdf/2405.20139, https://arxiv.org/pdf/2307.07697, and [2] focus on how to retrieve relevant subgraphs, we study the question of how to encode any given graph (whether retrieved or directly provided) for LLM consumption.
>
> FoGE provides a theoretically grounded and parameter-free encoding that could, in fact, enhance retrieval based approaches. Instead of textualizing retrieved subgraphs (commonly used), many of the approaches could use the Fock space encoding to preserve graph structure more faithfully. In this sense, our method is complementary to retrieval mechanisms — while they determine which graphs to process, FoGE shows how to encode them for the LLM.
>
> We have updated the manuscript to clarify this distinction (on page 1 - introduction), and remove potential confusion with retrieval-focused RAG approaches. We appreciate any additional suggestions for improvements.

---

### Author Response · Authors · 2024-11-21
**General response**

We would like to thank all the reviewers and express our gratitude for the time and effort they put into their reviews. We appreciate their constructive comments and we have already updated and uploaded a new version of the manuscript (the updates can be seen in blue color). Below, we will answer each reviewer’s comments separately and we remain available for any further questions.

---

### Author Response · Authors · 2024-11-28
**Kind Reminder: Rebuttal Period Ending Soon**

Dear reviewers,

We hope this message finds you well. We thank you again for your time and effort. We are writing to kindly remind you that the rebuttal period is nearing its conclusion. We have addressed your comments in detail and would be grateful for any further feedback or updates you may have before the deadline.

Thank you again for your time and thoughtful review of our work.

---

### Meta-Review · Area_Chair_5AHG · 2024-12-20

**Metareview:**

The paper proposes a parameter-free graph encoder inspired by Fock space to align graph embeddings with a frozen LLM via a linear adapter. While the concept of leveraging Fock space for graph representation is innovative, significant concerns regarding its experimental validation, imprecise statements and reproducibility make this work unsuitable for acceptance at this time. Specifically, the efficiency and effectiveness comparison with existing methods are not convincing. The baselines are insufficient, missing a wide range of LLM-as-assistant methods. The used datasets are also limited, lacking traditional graph datasets and graph foundation model settings.

**Additional Comments On Reviewer Discussion:**

1. Justification for Using LLMs in Graph Reasoning Tasks:
Point Raised: Reviewers questioned the necessity of using LLMs for traditional graph reasoning tasks that do not inherently involve textual data.
Authors' Response: The authors argued that LLMs are general-purpose reasoning engines, capable of handling non-textual modalities when appropriately encoded. They provided examples of multi-modal and non-textual applications of LLMs, such as for protein structures and tabular data.
Evaluation: While the argument was theoretically sound, it lacked experimental evidence specific to this work. The rationale remained unconvincing for traditional graph tasks where specialized methods already excel.
2. Efficiency Claims:
Point Raised: Reviewers were skeptical of the claimed efficiency benefits, highlighting that the computational cost is dominated by the LLM rather than the parameter-free graph encoder.
Authors' Response: The authors provided additional comparisons, reporting that FoGE-LLM required fewer epochs and exhibited marginal improvements in training and inference times compared to GraphLLM. They emphasized the ability to encode graphs offline as a potential advantage.
Evaluation: The additional data clarified some aspects but showed only minor efficiency gains. These improvements were not substantial enough to justify the claimed advantages in real-world settings, particularly given the dominance of LLM computation costs.
3. Performance on Specialized Tasks:
Point Raised: The proposed method often underperformed compared to GraphLLM on advanced graph reasoning tasks.
Authors' Response: The authors highlighted the simplicity of their approach and noted that their method performed competitively despite being task-agnostic and requiring fewer parameters. They argued that the versatility of FoGE outweighed the small performance gaps.
Evaluation: While the simplicity and versatility were acknowledged, the performance gap raised concerns about the practical utility of the method in scenarios where high accuracy is essential.
4. Comparison with Related Work:
Point Raised: Reviewers found the comparison with RAG-based methods and other graph prompting approaches unclear or misleading.
Authors' Response: The authors clarified that their work was complementary to RAG methods and emphasized distinctions between their parameter-free approach and trainable graph encoders. They updated the manuscript to better contextualize these differences.
Evaluation: The clarification improved the manuscript, but the initial confusion and lack of focused comparisons detracted from the work’s overall presentation.
5. Experimental Reproducibility:
Point Raised: Reviewers noted a lack of transparency in experimental settings, such as hyperparameters and prompt generation.
Authors' Response: The authors committed to releasing their code and added details on hyperparameters and prompts in the appendix.
Evaluation: The added details were appreciated, but the initial omission of this information affected the review process negatively.
6. Capturing High-Order Interactions:
Point Raised: Questions were raised about whether the proposed method could capture high-order interactions in graphs.
Authors' Response: The authors argued that the Fock-space formalism inherently captures high-order interactions and cited clustering results from protein datasets as evidence.
Evaluation: While the argument was conceptually valid, it would have been stronger with more direct experimental validation on complex graph tasks.

All the points contribute to my final decision.

---

### Decision · Program_Chairs · 2025-01-22

Reject